# Can Muscle Mass Be Maintained with A Simple Resistance Intervention in the Older People? A Cluster Randomized Controlled Trial in Thailand

**DOI:** 10.3390/ijerph19010140

**Published:** 2021-12-23

**Authors:** Bumnet Saengrut, Takeshi Yoda, Yumi Kimura, Yasuko Ishimoto, Rujee Rattanasathien, Tatsuya Saito, Kanlaya Chunjai, Kensaku Miyamoto, Kawin Sirimuengmoon, Rujirat Pudwan, Hironobu Katsuyama

**Affiliations:** 1Nursing Service Department, Maharaj Nakorn Chiang Mai Hospital, Faculty of Medicine, Chiang Mai University, Chiang Mai 50200, Thailand; bumnet_s@cmu.ac.th (B.S.); rujee.r@cmu.ac.th (R.R.); kanlaya.c@cmu.ac.th (K.C.); rujirat.pu@cmu.ac.th (R.P.); 2Department of Public Health, Kawasaki Medical School, Kurashiki 701-0192, Japan; katsu@med.kawasaki-m.ac.jp; 3Department of Health and Sports Science, Kawasaki University of Medical Welfare, Kurashiki 701-0193, Japan; i-yasu@mw.kawasaki-m.ac.jp; 4Graduate School of Human Sciences, Osaka University, Osaka 565-0871, Japan; 5Division of Medical Science in Sports and Exercise, Faculty of Medicine, Tottori University, Tottori 680-8550, Japan; t.saito@tottori-u.ac.jp; 6Faculty of Education, Kagawa University, Takamatsu 760-8522, Japan; kenmiyamoto@ed.kagawa-u.ac.jp; 7Faculty of Associated Medical Sciences, Chiang Mai University, Chiang Mai 50200, Thailand; kawin_sirimuengmoon@cmu.ac.th

**Keywords:** sarcopenia, simple resistance, older people, community-dwelling

## Abstract

The aging population is rapidly increasing worldwide. Sarcopenia is a common and important health problem among older people. The prevalence of sarcopenia among older Thai adults is increasing. Exercise intervention for sarcopenia prevention may significantly improve muscle strength, body balance, and muscle mass. Therefore, this study aimed to investigate the effects of a simple resistance intervention (SRI) program in preventing sarcopenia on physiological outcomes among community-dwelling older Thai adults. This study was a 12-week randomized controlled trial, which included 80 community-dwelling older adults in Chiang Mai, Thailand, who were randomly assigned into control (40 participants who performed usual exercise) and intervention (40 participants who performed the SRI program) groups. The SRI program was a home-based program consisting of 30 min of resistance exercise three times/week for 12 weeks, health education on sarcopenia. After 12 weeks, all physiological outcomes were measured and were significantly improved in the intervention group compared with baseline; hand grip, skeletal muscle mass index, and walking speed were significantly improved in the intervention group compared with the control group. Based on our results, the SRI program may prevent muscle weakness in community-dwelling older people in Thailand.

## 1. Introduction

The number of older people with sarcopenia or frailty increases with the growing number of older people worldwide. Sarcopenia is classically defined as a decrease in muscle mass, muscle strength, and physical function [1], but in recent years, further loss of muscle function has been added and interpreted [2]. Frailty is defined as a decrease in the body’s reserve capacity with age [3], making it more susceptible to health problems. It increases the risk of falls and results in poor quality of life and other diseases. Sarcopenia and frailty are important indicators of deteriorating health prognoses in older people [4] and are the main risk factors of diseases that significantly impair quality of life [4,5], such as falls [6,7,8,9] and associated femoral neck fractures [10], cerebral hemorrhage [11], and dementia [12]. Therefore, preventing sarcopenia and frailty is an urgent issue.

In many countries, measures to combat sarcopenia and frailty are becoming more important to prolong the period of time when people do not need assistance or care. Physical activity is one of the most important factors in the prevention of sarcopenia, and many researchers have found benefits from physical exercise in improving the functional capacity of frail older people [13,14,15,16,17,18]. Therefore, many researchers have been developing and publishing training methods that can be easily followed by older people. For example, Dunsky et al. studied the prevention of muscle weakness through exercise intervention using step aerobics and a stability ball [19], whereas Kanda et al. stated that slow-motion movement is more effective in preventing flail than regular exercise [20]. Syed-abdul et al. [21] and Baker et al. [22] tried community-based resistance training for eight weeks, respectively, and several studies have shown compelling evidence supporting the benefits of targeted physical activity programs for older adults to increase muscle strength [23,24,25,26,27,28]. 

These previous studies were all conducted with older people in developed countries, such as European countries, the United States, and Japan. In the tropics of Southeast Asia, where the population is rapidly aging, exercise interventions must be able to be sustained regardless of climatic conditions and without the use of complicated machines or systems. Therefore, we decided to simplify the exercise used in previous studies. We have developed a simple resistance intervention (SRI) program, which is easy to perform without using machines or special training kits, and allows training in any place; however, whether it helps older people to maintain muscle strength remains unclear. Thus, we conducted a randomized controlled trial to investigate the effect of the SRI program on muscle strength in a group of older people living in the suburbs of Chiang Mai, Thailand.

## 2. Materials and Methods

### 2.1. Ethical Approval and Funding Support

The study protocol was approved by the ethical committees of Kawasaki University of Medical Welfare (Approval number: 18-102) and Chiang Mai University (Approval number: NUR-2562-06120). This study was registered with UMIN-CTR (Registered number: UMIN000046042). All participants provided written informed consent to participate in the study. This research was funded by MEXT KAKENHI, grant number 19H04352 and Kawasaki University of Medical Welfare research grant, grant number H30-018.

### 2.2. Participants and Randomized Method

Participants were members of two community clubs for older adults, registered with the Nursing Department of Chiang Mai University, and had been surveyed in a previous study on community health [29,30]. The participants were older community-dwellers who could walk to the venue by themselves and understood the instructions and exercises. Although no actual cognitive function tests were conducted, both the intervention and control groups understood the contents of the program and were able to perform the measurements twice; thus, it was concluded that there were no problems with cognitive function. The cluster unit was the community group, where each participant was randomized into an intervention or control group. Because the exercise intervention was conducted within village sites, the subjects were randomized by cluster (i.e., village) to avoid contamination and enhance feasibility. Block randomization was used (block size = 2) to ensure that approximately equal numbers of community-dwellers were distributed. The group assignment coding (0 for control and 1 for intervention) was concealed from the research group. The assessment staff was blinded to the participant randomization assignments. Participants were explicitly informed regarding the group to which they were assigned and were reminded not to discuss their randomization assignment with the assessment staff. It was not possible to conceal the group assignment from the staff involved in the training.

Exclusion criteria were as follows: (1) systolic blood pressure 180 or higher and/or diastolic blood pressure 110 or higher in the first time measurement. (2) Already diagnosed and treated for dementia. (3) Those who could not walk on their own.

The sample size was calculated to be 73 in total and the difference in the mean skeletal muscle mass index (SMI) after 12 weeks between the intervention and control groups was set at 1.0, standard deviation at 1.5, significance level at 0.05, and power at 0.8.

### 2.3. Intervention Procedure

The content and methods of the SRI are shown in Figure 1. This exercise program was developed under the supervision of an exercise specialist as a training method that older people can easily perform. The SRI program consisted of three exercises, as follows: squat, heel lift, and thigh raising. These three exercises were called “resistance exercises” and are widely used by older people and athletes for strength training to increase muscle mass and improve muscle strength [31]. In particular, the chair-standing exercise can improve motor function in older people [32], and increases endurance through resistance exercise which can improve leg strength and walking ability in older people [33], making chair-standing an important exercise for maintaining muscle mass. SRI was performed 10 times a day, three times a week.

On the first day of the intervention, after baseline measurements were taken, the SRI program was introduced to the intervention group only, who were instructed to perform the exercises. All exercises were designed to prevent falls in older people when using a chair, squatting when getting up from the chair, and holding the back of the chair while raising the heels and thighs.

The first follow-up call was made for all participants in the intervention group 2 weeks after the initial exercise instructions. During the phone call, we asked each participant if he or she could continue the exercise and if there were any difficulties in continuing the exercise. Four weeks after the first instruction, the intervention group was asked to gather to confirm the implementation of the exercise and submit the implementation records. These follow-up calls and meetings were repeated every 2 weeks. Twelve weeks after the intervention, the two groups were collected, and a post-intervention survey was conducted (Figure 2).

### 2.4. Measurement Items

Participants in both groups responded to the initial survey using questionnaires on gender, age, annual income, education, marital status, number of family members living with them, history of drinking and smoking, exercise habits, and the type of exercise. Measurements were as follows: height, weight, body composition, blood pressure, hand grip, and walking speed for 6 m. Body composition was measured using a standing-posture 8-electrode multifrequency bio-impedance absorptiometry (BIA) analyzer (MC-780-A-N, Tanita, Tokyo, Japan); it had already been validated for estimating appendicular skeletal muscle mass by Yamada et al. [34] using a theoretical, age-independent equation. They established an age-independent multi-frequency-BIA equation for the Japanese population as follows: appendicular lean mass (ALM) = (0.6947 × (Ht2/Z50)) + (−55.24 × (Z250/Z5)) + (−10940 × (1/Z50)) + 51.33 for men, and ALM = (0.6144 × (Ht2/Z50)) + (−36.61 × (Z250/Z5)) + (−9332 × (1/Z50)) + 37.91 for women. In the cross-validation group, the equation had high determinant coefficients (R2 = 0.87 and 0.86, respectively). Blood pressure was measured with an automated sphygmomanometer (HEM-7130-HP, Omron, Kyoto, Japan). Each participant’s blood pressure was measured twice, once from each arm. Hand grip was measured using digital grip dynamometer (GRIP-D T.K.K.5401, Takei Scientific Instruments, Niigata, Japan) that was calibrated and designed to work in compression only, and was also measured twice for each arm with the participants in a standing position and the dynamometer resting at the side of the thigh. For the measurement of walking speed, the participants were asked to walk down a 10-meter corridor as if they were walking in daily life. From the time they reached the 2-meter marker from the start of walking, the measurer started a stopwatch and measured the number of seconds until they reached the 6-meter marker. These measurements were taken at baseline (pre-intervention period) and 12 weeks after the end of the intervention (post-intervention period), and the amount of change in each group was compared.

### 2.5. Statistical Analysis

Data at baseline and post-intervention periods were used in the analysis, which was measured twice, excluding the number of people who dropped out in the middle of the study (per protocol set). We considered this study to be more like the nature of an experimental trial than a pragmatic trial because we aimed to purely evaluate the effects of a simple exercise intervention. Thus, we selected protocol set analysis instead of intention to treat analysis by Schwartz et al. [35] The evaluators and data analysts were blinded to the study hypothesis and treatment allocation. We compared the differences in the mean values of the measurements, before and after the intervention, for the intervention and control groups. A two-factor (group and time) analysis of variance (ANOVA) with repeated measures was used. Leven’s homogeneity of variances tests were used before ANOVA was performed.

The analysis was repeated using baseline values as covariates. For each outcome variable, the level of significance corresponding to the main group (between subjects), time (within subjects), and interaction (group × time) effects was reported. A post hoc test was performed using Tukey–Kramer methods. Effect sizes were calculated as partial eta squared (η^2^) for the ANOVA results. We also performed a difference-in-differences analysis for the average of each measurement items between the two groups. Mann–Whitney’s U test was used to evaluate the average differences between two groups. The two-sided *t*-test and chi-squared test were used to evaluate continuous and categorical variables, respectively. Statistical significance was set at <0.05. JMP Pro 14.1.0 (SAS Institute Inc., Cary, NC, USA) was used for all analyses.

## 3. Results

As shown in Figure 2, the intervention and control groups were initially adjusted to have 40 participants each, but a total of 10 and 9 participants in the intervention and control groups respectively dropped out during the study period. Finally, the number of participants analyzed was 30 in the intervention group and 31 in the control group.

Reasons for dropping out for the intervention group were forgetting to do it three times a week (7 participants), moving to another district (1 participant), and not showing up (cancelled) on the post-intervention measurement (2 participants). The intervention group was cooperative with the fitness professional in charge of the intervention session.

No serious side effects or health problems resulting from the test session or the prescribed training session were observed among the participants in the intervention group. Socioeconomical background of all groups are shown in Table 1. In the two-way ANOVA, we found a significant group–time interaction in terms of average of hand grip (F = 9.14, *p* < 0.01, partial η^2^ = 0.216). For the intervention group, we observed significant improvement in body weight, hand grip, and SMI after intervention (*p* < 0.05) (Table 2).

Table 3 shows the results of the difference-in-differences analysis of the average of measurement items. The means of the differences between the pre- and post-intervention measurements were significantly higher in the intervention group than those in the control group for hand grip (control vs. intervention group; 0.22 vs. 1.55, *p* < 0.01) and SMI (0.18 vs. 0.21, *p* < 0.01, respectively). Walking speed was also significantly improved in the intervention group (control vs. intervention group; 0.003 vs. −0.023, *p* = 0.04, respectively).

## 4. Discussion

We found that 12 weeks of our simple exercise intervention led to weight loss, increased grip strength, and improved SMI. In particular, for grip strength, there was a significant difference in the interaction between intervention and time, suggesting that the continuation of the SRI program leads to an increase in this variable.

Although the exercise intervention in this study was mainly an approach to the muscles of the lower limbs, it is not designed to directly train grip strength it resulted in its improvement. Moreover, trunk training leads to an increase in grip strength [21]. In addition, the participants were told to hold the back of the chair to prevent from falling when doing squatting exercises and lower limb training, which may have indirectly strengthened their grip strength.

In our study, SMI was measured using the BIA method. SMI is sometimes measured using dual-energy X-ray absorptiometry (DXA) methods for a sarcopenia diagnosis. However, DXA is expensive, has limited portability, and requires radiation exposure. Compared with DXA, BIA is low-cost, easy to use, portable, and requires no radiation exposure [36,37,38]. Thus, we chose to use the Tanita MC-780-A-N to evaluate SMI. Tanita MC-780-A-N also measured about percentage of body fat. Although the percentage of body fat was not shown in the results because it was not main outcome of this study, it was significantly reduced in the intervention group compared to the control

The improvement in walking speed needs to be interpreted with caution. Two-way ANOVA showed no significant differences in the mean walking speed of the two groups, before and after the intervention. A difference-in-difference analysis, shown in Table 3, showed that the intervention group had a lower walking speed, *p* = 0.04, which was less than 0.05. Therefore, we can say that there is a significant difference. Since the training focused on the lower extremities, we expected the intervention group to improve their walking speed more, but the results showed that there was no significant difference, or, if we take the difference-in-difference, the walking speed was slightly lower. For gait, we used only walking speed from the sarcopenia diagnostic criteria, but it might have been better to use other scales, such as simple reaction time [39] or walking score [21].

The grip strength is not directly required for functional activities, such as walking, and it can distinguish older adults due to their motor abilities. Forrest et al. reported significantly lower grip strength in older adults who had physical limitations, such as standing up from a chair, walking, climbing stairs, and going out [40]. Zhang et al. found a low but significant relationship between grip strength and distance walked in a 6-minute walk test [41]. In several studies in which walking was judged to be slow (<0.80 m/s), grip strength thresholds for men ranged from 23.2 kg to 39.0 kg [42,43]. Sallinen found that grip strength thresholds of 37.0 kg for men and 21.0 kg for women were identified in older people who had difficulty walking 0.5 km or climbing stairs [44]. Grip strength has also been proven in studies as a biomarker of malnutrition. Ramírez-Vélez et al. reported that relative handgrip strength moderates the adverse effect of excess adiposity on dependence [45]. Examining a sample of older Chinese inpatients who were tested upon admission, Zhang et al. noted that those with lower grip strength were at greater risk of malnutrition, as measured by nutritional risk screening and subjective comprehensive assessment [46]. McNicholl found a significant association between grip strength and nutritional status, but indicated that grip strength “has little validity as a single nutritional indicator” [47]. Since more tested patients completed the grip strength assessment (92%) than the 5-m walk assessment (43%), they concluded that grip strength is "a more useful functional indicator" than the 5-m walk test [47]. Furthermore, a number of studies have noted covariation between grip strength and cognition, depression, and sleep [48,49].

Thus, it is assumed that grip strength measurement is an indicator that reveals not only overall muscle strength but also various other health conditions of older people. The significant increase in grip strength in the intervention group in the current study is thought to have improved the various health indicators of older people. However, clinicians and scientists should be cautious in using grip strength as a measure of overall strength, because it is not necessarily an indicator of overall fitness. This is because grip strength does not reflect overall strength [50], and there is evidence that it may give a better indication of overall strength when used in conjunction with muscle strength measurements of the lower extremities [51].

In our study, in addition to grip strength, we also assessed limb muscle mass using SMI, which includes the lower extremities, and found that SMI increased with the implementation of SRI compared to the control group. However, the results of two-way ANOVA did not show a significant difference. Therefore, this study cannot conclude whether SET improves SMI, and further research is needed to clarify this.

For people who usually do strength training or high-intensity work, SRI is too easy and is boring training. On the other hand, for people who usually live a sedentary life and hardly move, SRI may be painful. Almost all participants (96.7%) in the intervention group answered "yes" to questions about whether they exercise on a regular basis, indicating that they were not afraid to exercise. The control group without SRI also had a very high percentage of 90.3% who answered that they exercise regularly, although the results after 12 weeks were significantly better in the intervention group. Therefore, the addition of SRI to regular exercise may lead to a more effective prevention of sarcopenia. Sometimes, fast-intended-velocity resistance training may elicit greater improvements in functional capacity when compared to moderate-velocity resistance training [52,53]. However, a recent systematic review and meta-analysis suggests that there is inconclusive evidence to support the superiority of fast-intended-velocity resistance training to improve functional capacity when compared to moderate-velocity resistance training [54]. Our SRI program was also moderate-velocity resistance training, which was more acceptable and easier for older people living in developing countries.

Our study has the following limitations. First, although the participants were randomly selected, the sample size was smaller than the original calculated sample size, so we cannot eliminate the possibility of bias in the results. In addition, it was difficult to generalize our study results for other settings or populations. Second, we applied PPS method instead of ITT. This method was chosen in order to better ascertain the nature of the experiment, but because dropouts were not included in the analysis, the interpretation of the results was limited. Furthermore, in order to reach the ideal sample size for this study, we recruited more participants than the calculated number, but, as a result, we were not able to reach the ideal sample size because there were more dropouts than the calculated number. In other words, caution should be exercised when standardizing the results. Third, we did not set lower extremity muscle mass as an outcome, despite the fact that we did strength training of the lower extremities. This was because we hypothesized that simple lower extremity training alone would lead to an increase in grip strength, which is a diagnostic criterion for sarcopenia. However, because we used standing-posture 8-electrode multifrequency BIA analyzer, we could directly compare lower extremity muscle mass alone. The results were recorded, and if ethical issues can be cleared, we would like to continue this study in the future. Lastly, owing to the nature of the intervention, participants who received the SRI program could not be blinded.

Despite the above limitations, our study revealed that SRI has certain effects in preventing muscle weakness in older people in Chiang Mai, Thailand. Active aging is very important for the aging population in Southeast Asia, which will continue to increase in the future. Therefore, we hope that the use and continuation of the SRI program will help to prevent sarcopenia in older people.

## 5. Conclusions

The current study examined whether the SRI program can help prevent muscle weakness and improve motor function of older people living in a community in Thailand. As a result, measurements taken 12 weeks after intervention showed that grip strength and SMI improved significantly compared to the control group, and in particular, grip strength showed significant differences in both time and group interaction. Therefore, the SRI program may prevent muscle weakness in community-dwelling older people in Thailand. Furthermore, we believe that the continued implementation of such easy-to-follow exercises will be important in preventing sarcopenia in the future.

## Figures and Tables

**Figure 1 ijerph-19-00140-f001:**
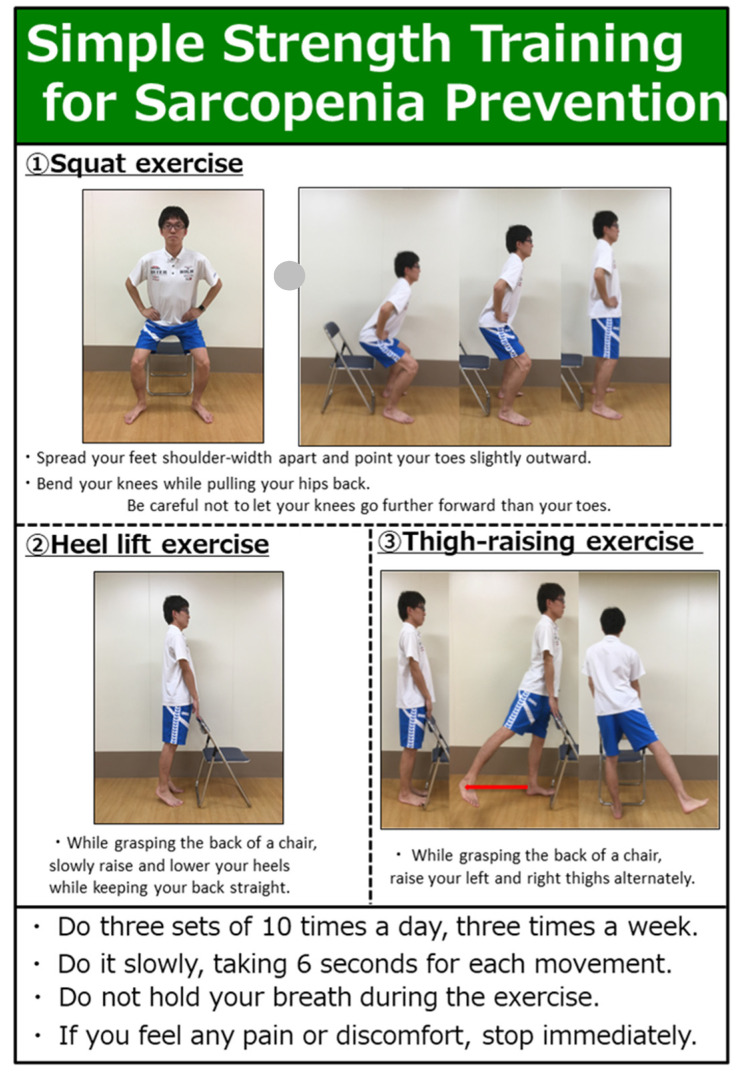
Contents of SRI.

**Figure 2 ijerph-19-00140-f002:**
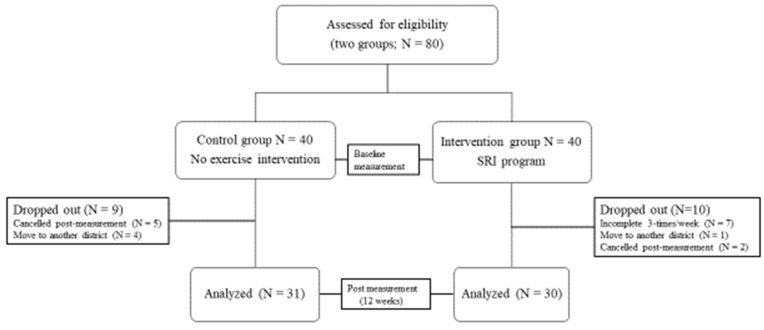
Flow diagram of participants.

**Table 1 ijerph-19-00140-t001:** Baseline characteristics of participants (*n* = 61).

Variables	Control Group *n* = 31	Intervention Group *n* = 30	*p* ^1^
Gender (%)			
Male	11 (35.5)	7 (23.3)	0.40
Female	20 (64.5)	23 (76.7)	
Average age (SD)	68.7 (6.64)	67.6 (4.82)	0.46
Average annual income, Thai baht (SD)	75,806.4 (142850.2)	46,766.6 (90319.1)	0.35
Education (%)			
Primary school	18 (58.1)	16 (53.3)	0.34
Secondary school	1 (3.2)	5 (16.7)	
High School	3 (9.7)	4 (13.3)	
College	2 (6.5)	2 (6.7)	
Bachelor’s degree	7 (22.6)	3 (10.0)	
Marital Status (%)			
Single/widow	12 (38.7)	13 (43.3)	0.80
Married	19 (61.3)	17 (56.7)	
Number of family members (%)			
1	2 (6.5)	5 (16.7)	0.32
2	8 (25.8)	8 (26.7)	
3	4 (12.9)	7 (23.3)	
4	10 (32.3)	4 (13.3)	
5+	7 (22.6)	6 (20.0)	
Smoking (%)			
Current smoker	3(9.7)	0	0.06
Quit/ Non-smoker	28 (90.3)	30 (100.0)	
Alcohol (%)			
Drink alcohol	9 (29.0)	6 (20.0)	0.55
Never	22 (71.0)	24 (80.0)	
Regular Exercise (%)			
Yes	28 (90.3)	29 (96.7)	0.61
No	3 (9.7)	1 (3.3)	
Average body weight, kg (SD)	61.8 (9.6)	57.0 (10.8)	0.07
Average height, cm (SD)	157.8 (1.43)	153.9 (1.45)	0.05
Average BMI (SD)	24.9 (4.22)	24.0 (3.96)	0.41
Average systolic blood pressure, mmHg (SD)	131.1 (13.7)	133.2 (14.9)	0.57
Average diastolic blood pressure, mmHg (SD)	71.5 (1.34)	76.3 (1.36)	0.01

^1^ *p* was the results of chi-squared test for categorical variables and two-sided *t*-test for continuous variables, respectively. SD, standard deviation.

**Table 2 ijerph-19-00140-t002:** Comparisons of changes in the intervention and control groups.

Measurement Items	Control Group (Average)		Intervention Group (Average)		Two-Way ANOVA			
	Pre	Post	Pre	Post		F	*p*	Effect Size
Weight (kg)					Interaction	0.95	0.33	0.016
	61.8	61.5	56.9	56.3 *	Group	3.74	0.05	0.06
					Time	5.17	0.02	0.081
Hand grip (kg)					Interaction	9.14	<0.01	0.216
	24.4	24.6	23.8	25.4 *^,#^	Group	0.0004	0.98	<0.001
					Time	16.28	<0.01	0.134
SMI (kg/m^2^)					Interaction	0.07	0.78	0.001
	6.80	6.98	6.47	6.68 *	Group	2.61	0.11	0.042
					Time	11.82	<0.01	0.167
Walking speed (m/s)					Interaction	2.03	0.16	0.033
	0.63	0.64	0.64	0.62	Group	0.05	0.81	0.001
					Time	1.13	0.29	0.019

* Significantly different from the pre-intervention time point (*p* < 0.05); ^#^ significantly difference from the control group (*p* < 0.05); SMI, skeletal muscle mass index; effect size was shown as partial η^2.^

**Table 3 ijerph-19-00140-t003:** Comparisons of average differences between pre- and post-outcome variables in the control and intervention groups.

	Average Differences between Pre-and Post-		*p* ^1^
	Control Group (*n* = 31)	Intervention Group (*n* = 30)	
Average hand grip (kg)	0.22	1.55	<0.01
Average walking speed (m/s)	0.003	−0.023	0.04
Average SMI (kg/m^2^)	0.18	0.21	<0.01

^1^ Mann–Whitney U test; SMI, skeletal muscle mass index.

## Data Availability

Data presented in this study are available upon request from the corresponding authors (T.Y., Y.K.). Data are not publicly available due to privacy concerns.

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
