# Peer review of "Can Muscle Mass Be Maintained with A Simple Resistance Intervention in the Older People? A Cluster Randomized Controlled Trial in Thailand"

_ijerph, 2021, doi:10.3390/ijerph19010140_

Round 1

Reviewer 1 Report

Dear authors,

The manuscript's idea is very interesting. However, I identified several major flaws during my review, considering the RCT design. You may see my comments in the attached file.

Regards.

Author Response

Thank you very much for your careful review. Please refer to the attached file for comments on the review.

Reviewer 2 Report

Interesting study conducted by Saengrut et al. I congratulate the authors for completing such a wonderful study. The manuscript must go through professional writing service to improve writing style. The introduction and discussion sections needs much more work. I have suggested some references that the authors can use to improve these sections.

Overall, the study is excellent and can be published once these changes are made.

Author Response

(The authors gave the same response as above.)

Reviewer 3 Report

It seems to me an interesting paper, easy to read and with a simple methodology and of great use and applicability.

I suggest a few changes and/or comments in the attached file.

Author Response

(The authors gave the same response as above.)

Round 2

Reviewer 1 Report

Dear authors, 

There are some issues not correctly addressed in your reviewed version.

Point 3: You finished the protocol with 61 participants, not considering the drop-outs. Was the power analysis recalculated to ensure the minimal 0.8 power? If not, explain how the results can be extrapolated to other people.

Response 3: Thank you for your comments. We considered drop-out, so we recruited 80 people in anticipation of dropouts, but there were more dropouts than we expected, so the calculated sample size was not met. However, it is inappropriate as a research method to recalculate and revise the sample size. We honestly presented the calculated sample size and conducted the analysis based on the data obtained. We have added a note in the limitations of the study section that caution should be exercised in interpreting the results (line 341-347).

2nd review: This is not a limitation, but an impairment to extrapolate the results. If a power analysis is not met due to drop-out, the results are fully limited to the sample. This means that the study or the technique does not provide enough adherence, which impairs its practical usefulness. And sure you cannot recalculate the sample size, but instead you must perform an intention to treat analysis to consider that effect on the results.

Point 9: (line 131-135) Any asumption check (normality, at least?)?  Did you use a mixed factorial ANOVA? The interaction was not assessed? Why not? Did you use any post hoc correction? Explain. For which purpose did you use t-tests? This is very confusing. Please, restructure. Why didnt you perform an effect size analysis?

Response 9: Thank you for your pointed-out. We were missing about information of two-way ANOVA, so we added the details of statistical methods (line 185-192).

2nd review: I could not notice any normality test. Please, make a statement for that issue.

Point 12: (line 146-148) This configures a per protocol analysis, and the recommended analysis for a RCT is the intention-to-treat. Please, justify why you used the current method or change the analysis, results, etc.

Response 12: Thank you for your comments. We considered this study to be more in the nature of an experimental trial than a pragmatic trial because we aimed to purely evaluate the effects of a simple exercise intervention. Thus we selected per protocol set analysis instead of intention to treat analysis by Schwartz et al. We added this sentences into the methods section (line 179-182) and limitation part in the discussion section (line 341-344).

2nd review: This does not make any sense. You had many people leaving the intervention (>20%), you are testing the effect of an intervention and you classified your study as a RCT. Please, refer to CONSORT guidelines where it strongly recommends the ITT analysis.

Author Response

Dear Reviewer 1, 

We thought we addressed all of your questions appropriately, but it seems that some of them were incomplete. Our apologies.

Point 3: You finished the protocol with 61 participants, not considering the drop-outs. Was the power analysis recalculated to ensure the minimal 0.8 power? If not, explain how the results can be extrapolated to other people.

Response 3: Thank you for your comments. We considered drop-out, so we recruited 80 people in anticipation of dropouts, but there were more dropouts than we expected, so the calculated sample size was not met. However, it is inappropriate as a research method to recalculate and revise the sample size. We honestly presented the calculated sample size and conducted the analysis based on the data obtained. We have added a note in the limitations of the study section that caution should be exercised in interpreting the results (line 341-347).

2nd review: This is not a limitation, but an impairment to extrapolate the results. If a power analysis is not met due to drop-out, the results are fully limited to the sample. This means that the study or the technique does not provide enough adherence, which impairs its practical usefulness. And sure you cannot recalculate the sample size, but instead you must perform an intention to treat analysis to consider that effect on the results.

2nd Response: Thanks for your comment. You are right, the ITT will show the practical EFFECTIVENESS of this intervention. However, it’s not that the results cannot be completely extrapolated, even if the PPS excludes the dropouts. The analysis excluding dropouts will show the true EFFICACY of the intervention. In any case, due to the small sample size of this intervention study, we plan to continue this study in the future. Also, at that time, we will conduct the analysis in ITT as you pointed out.

Point 9: (line 131-135) Any asumption check (normality, at least?)?  Did you use a mixed factorial ANOVA? The interaction was not assessed? Why not? Did you use any post hoc correction? Explain. For which purpose did you use t-tests? This is very confusing. Please, restructure. Why didnt you perform an effect size analysis?

Response 9: Thank you for your pointed-out. We were missing about information of two-way ANOVA, so we added the details of statistical methods (line 185-192).

2nd review: I could not notice any normality test. Please, make a statement for that issue.

2nd Response: Thanks for your comment. We confirmed normality test using normal Q-Q (Quantile-Quantile) plot. Since the data were generally distributed in a straight line, it was determined that there was normality.

Point 12: (line 146-148) This configures a per protocol analysis, and the recommended analysis for a RCT is the intention-to-treat. Please, justify why you used the current method or change the analysis, results, etc.

Response 12: Thank you for your comments. We considered this study to be more in the nature of an experimental trial than a pragmatic trial because we aimed to purely evaluate the effects of a simple exercise intervention. Thus we selected per protocol set analysis instead of intention to treat analysis by Schwartz et al. We added this sentences into the methods section (line 179-182) and limitation part in the discussion section (line 341-344).

2nd review: This does not make any sense. You had many people leaving the intervention (>20%), you are testing the effect of an intervention and you classified your study as a RCT. Please, refer to CONSORT guidelines where it strongly recommends the ITT analysis.

2nd Response: Thanks for pointing this out. The guidelines recommend ITT, but it is not mandatory. We did not do an ITT analysis because we wanted to examine the effects of the exercise itself in this study. Do you understand the implications of this? We guess what you are saying is that we should examine practical EFFECTIVENESS, but that is not our purpose. By the way, this was important point and still weak point of our study, so we explained these points carefully in the limitations.

Reviewer 2 Report

The authors have well addressed my concerns. I have no more comments. I congratulate the authors for completing such a wonderful study.

Reviewer 3 Report

Good job